# Voices from the Margins: Aotearoa/New Zealand Young Carers Reflect on Their Experiences

**DOI:** 10.3390/ijerph20156511

**Published:** 2023-08-03

**Authors:** Lauren Donnan, Janet S. Gaffney, Toni Bruce

**Affiliations:** 1Faculty of Arts, Waipapa Taumata Rau University of Auckland, Auckland 1023, New Zealand; 2School of Curriculum and Pedagogy, Faculty of Education and Social Work, Waipapa Taumata Rau University of Auckland, Auckland 1023, New Zealand; janet.gaffney@auckland.ac.nz (J.S.G.); t.bruce@auckland.ac.nz (T.B.)

**Keywords:** young carers, Aotearoa/New Zealand, culture, holistic wellbeing, hidden, overwhelm

## Abstract

Young carers are a largely invisible and unsupported population of Aotearoa New Zealand (NZ) children and youth aged 25 years and under who physically, emotionally, socially, and/or spiritually support loved ones experiencing ill health, disability, substance misuse, mental illness, or advanced age. The total number of young carers in NZ is unknown because census data only capture those aged 15–25. The nine published NZ studies recommend further research, policies, and services for young carers. However, there is a lack of young carer voices to inform their implementation. This paper provides insight into the experiences and needs of 28 young carers, the largest self-identified cohort in NZ research. Throughout phenomenographic interviews, young caring was described as a natural and valued part of being in a whānau/aiga/family, aimed at supporting their loved one’s holistic wellbeing. However, the overwhelming nature of caring without recognition or support resulted in poor educational, social, and mental health outcomes for young carers. This paper concludes with a contextualised NZ young carer definition and an urgent call to address the Carers’ Strategy Action Plan and listen to, and deliver on, young carers’ expressed needs.

## 1. Introduction and Literature Review

Census data identify over 40,000 young carers in Aotearoa/New Zealand (NZ) (Aotearoa is the Māori term for New Zealand (http://www.maori.com/aotearoa (accessed on 3 April 2023)). Te reo Māori (Māori language) is interspersed throughout this paper to acknowledge Māori as tangata whenua (indigenous people) [1]) aged 15 to 25 years [2]. This number is a significant underrepresentation as young carers can begin their roles as young as three years old [3], and while caregiving is conceptualised differently by different individuals and whānau/aiga/family (whānau), care is often a natural part of being in a whānau unit [4]. (Whānau is the Māori term and aiga is the Pacific term for the family unit. The phrase whānau/aiga/family shortened to whānau after the first use recognises the diversity of family groups and understandings of caregiving represented in NZ. It is used here in its wider connotation of extended family, unless otherwise specified.) Secondary analysis of NZ census data of young carers aged 15–25 found that they were most often female, from low socio-economic or high deprivation areas, and identified as being of Pākehā (a common term for white New Zealanders) (64%), Māori (29%), or Pacific region heritage (16%) [2]. (The term Pacific refers to individuals living in NZ who have ancestral/heritage links to the Pacific region.) Despite young carers comprising 10% of the total NZ carer population [5], young carers’ voices and experiences are almost invisible in the broader caring literature. Nine publications address NZ young carers, with four incorporating original data sets [6,7,8,9,10,11,12,13,14]. A key theme among NZ studies is the hidden and unsupported nature of young carers [11], who are described as “a vulnerable, invisible group who require recognition and respect” [10] (p. 7). The difficulty of recruiting NZ young carer participants is another significant theme, alongside the need for more research [14]. The potential for research to inform policy and service outcomes for young carers is also stressed [9,10].

The small body of NZ young caring scholarship is complemented by the larger corpus of international studies, which provide insight into young caring. While young carers represent a diversity of children and youth in a range of caring relationships, young carers are most often school-aged [15], females [16], caring for mothers [17] and whānau members with a disability [18]. Children and youth appear more likely to take on care if they are the eldest in their whānau [19] and living in households of low socioeconomic status or with a sole parent [20]. Motivations for young caring include a lack of services for individuals with disabilities and illnesses and their families [13], resistance to services due to disability-related stigma and financial barriers [20], cultural and familial preferences for within-whānau care [18], and young people’s desires to undertake care [10]. The hidden or invisible nature of young caring means that identification poses a significant challenge in all parts of the world [7,21].

While no consistent definition of young caring exists, the key factor distinguishing young carers’ roles is the regularity or substantial level of tasks carried out [18,22]. Care tasks include household chores, emotional and social support, intimate care, and medical or nursing support [16,23]. While young carers’ enjoyment of providing care is highlighted [13,24], this benefit is outweighed by negative educational, socialisation, and health impacts including early school exit, loss of opportunities to socialise with peers, and poor mental health [10,25].

The recent global expansion of international young-carer projects highlights inconsistencies in services both within and between countries [10,20,24], varying from policy and legislative support to the absence of any recognition or support [26]. However, even when policy change is implemented to meet the needs of young carers, turning policy into action can be slow [16,27].

A lack of urgency translating young carer-related policy into action is evident in NZ. The Carers’ Strategy and related Action Plans recognise carers aged under 25 years as one of four target populations (alongside Māori, Pacific, and older carers) who require recognition and support over and above the general carer population [5,28]. The most recent Carers’ Strategy Action Plan 2019–2023 outlines four young carer-related Actions that, if effectively implemented, will result in young carers being identified, their voices being heard, and supports being provided based on their expressed needs [5]. However, in the five years that the Action Plan has been in effect, none of the four young carer Actions have been achieved. NZ’s limited policy response is reflected in its classification as a level four “preliminary” country for young carer responsiveness on a scale of 1–7, where 1 is young carers’ full recognition in policy and 7 is no recognition at all [26] (p. 752). Consequently, NZ young carers remain a “silent, taken-for-granted, population” whose needs require urgent attention and address [10] (p. 7).

This article provides insight into the expressed experiences and needs of NZ young carers with the aim to translate their voices into research, policy, and service implementations. It is timely given the recent release of the 2023 Commonwealth Charter for Young Carers [29] urging countries currently or formerly associated with the British Empire to uphold and protect the rights of young carers, and the NZ Carers’ Strategy Action Plan coming up for review.

## 2. Methodology and Methods

With a Bioecological Systems Theory (BST) underpinning, this research sought to discover innovative and relevant knowledge to influence policy and practice [30,31]. BST facilitated a holistic insight into young carers’ experiences at five levels: the microsystem of immediate settings (e.g., home, classroom); the mesosystem of interconnections between settings (e.g., between a young carer’s home, school, and community groups); the exosystem in which young carers are not active participants but events occur that affect or are affected by what happens in young carer settings (e.g., parents’ workplaces); the macrosystem, which is a particular setting’s “blueprint” [32] (p. 26); and the chronosystem of environmental changes occurring over the life course [30,31].

Phenomenography facilitated examination of the qualitatively different ways that participants understood their caring roles and their wider lives outside of care [33] (p. 335), enabling a focus on young carers’ ideas and experiences of the world [33], and attention to qualitative variations within and between participants’ experiences [34].

Participant recruitment. Recruitment aimed to identify current and former young carers who were diverse in gender, ethnicity, ages when caring, disability or illness of care recipient, relationship to care recipient, and geographical locale. Current young carers (≤25 years) could provide perspectives on their present-day immersion in the realities of being a carer, and former young carers could provide retrospective perspectives, thus allowing experiences from a wider timeframe to be included. Criteria for involvement were intentionally broad, and included individuals who currently or previously provided care when they were ≤25 years for at least one year. Care was defined as providing significant support for a loved one who had a disability, illness, mental illness, substance misuse, or who was elderly.

Given the known difficulties in NZ of recruiting young carers for research [7,10,12], a range of innovative, qualitative methods were used, including direct recruitment of young carers, rather than going through gatekeepers such as parents/caregivers or service providers because youth do not always disclose their roles to services [10,16] and cautious “gatekeepers” could have limited the study’s scope due to perceptions of the research as being sensitive [35] (p. 504).

Initial recruitment attempts were challenging. Four posters—each targeting a different age group—posted physically and online at universities, youth and disability organisations, and in the community failed to attract any participants. A story in a NZ carers’ magazine [36], written by the first author sharing her own young caring experiences, was largely unsuccessful, with only three young carers being recruited in a six-month period. However, a subsequent video interview with the first author about the discovery of her young caring identity in her early twenties after many years of caring in isolation, her experiences of caring, and the research, was shared widely on social media by over 100 individuals and organisations both within and well beyond care-related communities. This innovative recruitment method was successful, attracting interest from a further 25 individuals, many of whom had neither previously heard the term young carer nor identified as a young carer. Ultimately, 4 current and 24 former young carers took part in the study, most of whom had not previously self-identified as young carers. Participants represent the largest-known cohort of self-identified NZ young carers, incorporating diversity of ethnicity, age when caring, relationship to care recipient, and care recipient disability or illness (see Table A1, Appendix A).

Despite being a non-random sample, the participants were broadly representative of the gender, age, ethnicity, and current/former status identified in other NZ studies [6,7,11,23]. In line with existing NZ studies, current young carers were difficult to recruit [8,10], with only 4 participants (14%) being current young carers and the remaining 24 (86%) being former young carers. The gender breakdown also reflected existing research with majority being female (n = 25, 89%) [17,23]. Participants identified with six different ethnicities: Pākeha, Māori, three Pacific heritages, and Asian. Although the majority identified with only one ethnicity, one identified with two ethnicities and one with three ethnicities. The most common was Pākeha (83%), followed by Maori (25%), Pacific heritages (17%), and Asian (4%).

In Table A1 (see Appendix A), the care recipients’ disabilities or illnesses appear in five main categories to ensure confidentiality. Neurological includes conditions such as autism, epilepsy, multiple sclerosis, ADHD, cerebral palsy, dementia, and muscular dystrophy, although some of these also manifest in physical disability. Physical includes paraplegia, arthritis, elderly frailty, and unidentified physical disability. Mental includes bipolar disorder, paranoid schizophrenia, depression, and unidentified mental illness. Medical includes cancer, stroke, chronic illness, diabetes, heart attack, and encephalitis. Substance includes drug misuse and abuse. Table A1 shows that 60% (n = 15) of young carers were providing care that encompassed multiple categories of illness or disability.

Informed consent. Informed consent was obtained through Participant Information Sheets and Consent/Assent Forms tailored to participants’ ages, and a summary of the research was reiterated at the beginning of interviews. Participants were made aware of their right to withdraw from the research without consequence until they approved their transcript. Written consent was given by adult participants. Participants under 16 gave written assent together with parent/caregiver consent. Participants received information about organisations that could assist if their interviews conjured negative and upsetting feelings, and a researcher and participant plan was made for instances where physically or emotionally high-risk scenarios were shared by participants [37]. All identifying information was removed from transcripts and pseudonyms were assigned to participants.

Phenomenographic interviews. In-person interview locations were decided in collaboration with participants, and Skype interviews were conducted with participants who lived overseas or outside Auckland. Phenomenographic interviews have no fixed protocol [38], so the only prearranged element was several questions asked to open the discussion about young caring and used throughout interviews to bring attention back to the topic. The overarching research question was: What does it mean to be a young carer in NZ? Sub-questions explored young carers’ motivations, perceptions of their carer identities, impacts of caring in childhood, and access to formal supports.

Interviews centred on participants’ voices, with the order and sequence of topics driven by interviewees [39]. A natural conversational flow allowed participants to make and remake meaning of their often previously unaddressed experiences. Prompts were used to request clarification or elicit further elaborations. All interviews extended beyond the proposed sixty-minute duration, with some extending to three hours, to allow participants to fully explore their experiences.

Phenomenographic whole of transcript analysis. Audio-recorded interviews were transcribed verbatim and included field notes about body language and emotional responses [40]. Complete transcripts minus field notes were emailed to the participant, with a plain-language summary sent to younger participants alongside their full script. Participants could then respond, highlighting areas to expand upon or remove. Only minor amendments were made, and all participants approved their transcripts. No one withdrew.

Analysis utilised a whole of transcript approach, moving backwards and forwards through entire transcripts [41], interpreting participants’ conceptions of care within the wider context of their interview [38]. Throughout iterative transcript readings, participants’ experiences of care were compared and contrasted, and similar conceptions were grouped into “categories of description” [39] (p. 43), which were modified, adjusted, and deleted until a stable system encompassing all interviews was established [38]. Phenomenographic interjudge reliability was employed wherein the first author’s supervisors played devil’s advocates, requiring the first author to justify emerging categories [38]. NVivo software facilitated analysis with categories being housed in “nodes”, and quotes within each node remaining embedded within their interview context. NVivo also facilitated connections across interviews via text and word frequency. Overall, this approach enabled identification of the themes that formed the basis of the findings.

## 3. Findings

The findings present valuable insights into how the young carers conceptualised their roles in the context of their lived experiences in three key areas: the nature and extent of care they undertook, the support they and their whānau received, and the outcomes (educational, socialisation, and health) of providing care. Throughout, participants are identified by their pseudonyms. Other details—care status, gender, ethnicity, age, years spent caring, care recipient relationship, and disability or illness—are included where relevant and are judged to not risk an individual’s identity.

### 3.1. Variations in Onset of Young Caring Roles

While 30% (n = 8) of participants identified the sudden onset of care because of an accident or change in whānau dynamics, a gradual onset of care was most common, with over 70% (n = 20) stating that they *‘drifted’* or *‘crept into’* their roles. Former young carer Ngākau explained the sudden onset of caring for her siblings due to their mother’s mental illness and both parents working, stating; *‘I don’t think there was any transition [to] me caring; all I can remember is that I came home one day, and I was doing it’* (carer from 14 to 17 years old). Gradual care onset often reflected the steady progression of a care recipient’s disability or illness, alongside young carers’ increasing age and understanding of their care recipient’s needs. Former young carer Dan said that the onset of caring for his mother with mental illness *‘didn’t just happen overnight … there was a transition period … For her, things got worse and worse, but I [got] very adept at dealing with it’* (carer age 9–14).

Often participants described both gradual and sudden care onset within the one role, making it difficult to pinpoint the commencement of caring. For instance, care could gradually onset as a care recipient’s illness slowly progressed, then suddenly increase due to an abrupt health decline. Former young carer Chun explained that her mother ‘*had [medical condition] and we were already doing [care]’,* but when her mother later also developed a neurological condition, *‘that’s the turning point … when the carer role really begins’* (carer age 21–25).

Another reason role onset was difficult to pinpoint was the normalcy of children and youth providing care. Participants described wanting to carry out their natural whānau caregiving roles, identifying the *‘honour’*, *‘privilege’*, or *‘pleasure’* of enacting care. Former young carer Alice said that caring for her brother with a neurological condition *‘is a natural part of our family … we all help each other in some way, and so if I wasn’t having a role in helping [my brother], then I would feel I suppose more disconnected from him and the family’* (carer age 15–25). Conceptualising care as *‘natural’* was particularly common among Māori, Pacific, and Asian young carers who identified the cultural importance of enacting a collective model of care. Former young carer Kahurangi, who supported her father through his medical and mental health conditions, said that ‘*Māori have become so disempowered that they think authorities and services know more than them and we have compromised our cultural whanaungatanga way [of caring] and that’s a sad thing … That’s why I nursed my dad’* (carer age 13–25). [“*Whanaungatanga”* refers to a sense of family connection through shared experiences and working together [42].]

### 3.2. The Challenging and Complex Nature of Holistic Care

Young carers provided holistic care to meet multiple aspects of their care recipient’s wellbeing, comprising physical, emotional, social, and spiritual support. All young carers faced significant challenges carrying out their care task, the variation, type, and degree depended on the young carers’ age at the time of providing care, their relationship to the care recipient, and their care recipient’s disability or illness. (This manuscript discusses major themes across the sample, so it does not explicitly address age-related differences.) Such challenges created layers of complexity within young carers’ tasks that, while having a significant impact on how young carers experienced providing care, remained largely invisible throughout their time young caring. The phenomenographic interviews with a diverse cohort of former and current young carers were integral to identifying such complexities underlying their roles, allowing participants space to unravel their experiences and make and remake meaning of their timecaregiving. In particular, former young carers were able to not only reflect on the challenges underlying their roles but also explore how those challenges shifted over time, revealing complexities that current young carers who were embedded in their caring roles often could not recognise. The benefit of a diverse participant sample also provided insight into the variance in challenges experienced by young carers.

#### 3.2.1. Dimensions of Physical Care

Care tasks that addressed their loved ones’ physical welfare dominated participants’ initial discussions of care and included domestic, intimate, nursing, mobility, supervision, childcare, financial, service coordination, and translation tasks. All participants carried out domestic care encompassing cooking, cleaning, and food shopping. Many young carers conceived that domestic chores signalled the commencement of caregiving, but all young carers’ roles soon extended beyond domestic tasks.

Intimate care was carried out by 86% (n = 24) of young carers and included dressing, grooming, toileting, and bathing. Intimate care was often described as ‘*difficult*’ or ‘*embarrassing*’, especially in the many cases where care recipients were older than the young carers. Former young carer Miharo explained that caring for her mother with a physical and mental illness meant that *‘we had to hold mirrors while [mum] catheterised herself … I just didn’t think it was right … especially my brothers because it’s like ‘if I don’t like looking at it, how are you poor boys doing?’ And like, poor [mum], you could tell she didn’t like it as much as we didn’t like it, but it had to be done’* (carer age 9–16).

Nursing care was undertaken by 71% (n = 20) of participants and ranged from applying dressings to sores and wounds to spoon-feeding, administering or injecting medications, and checking vital signs. Nursing support appeared to intensify during palliative care and was the task most associated with young carers feeling unqualified. Former young carer Lucy said that she would be ‘*constantly changing the syringe on the pain pump …and taking blood pressure’* for her mother with a medical condition, *‘which I really wasn’t qualified to do’* (carer age 15–19).

Over 65% (n = 19) of young carers carried out mobility care; lifting care recipients in and out of wheelchairs or mobility vehicles, turning them in bed, or assisting with transferring from one setting to another. Participants said a lack of equipment made mobility care difficult. Former young carer Miharo explained that because her mother did not have a wheelchair, they *‘somehow had to transport her from her bed, and she was a big woman. [We] did it by putting her on a sheet, lifting her off her bed, putting her on the skateboard, and carting her around like that.’*

While all participants cited supervision or *‘watching over’* their care recipients as a key care task, supervision was a particularly significant role for young carers whose care recipients had a mental illness or substance misuse. Former young carer Melanie described *‘the biggest aspect of caring’* for her friend with a mental illness as *‘always watching for self-harm or expressing that he wanted to hurt people’* (carer age 19–21).

Childcare was undertaken by over 30% (n = 9) of participants and, in most cases, was necessitated due to parental illness or disability, especially in sole-parent households. Former young carer Fleur said that because her mother with a neurological condition *‘wasn’t very well, I would do things like get up in the morning and look after my little brother … I was six years old and changing nappies and making bottles and doing quite a lot around the house’* (carer age 6–25).

Financial care was carried out by 68% (n = 19) of young carers who managed the whānau finances, including 10 participants as young as 12 years old who worked as a ‘financial contribution’ to their whānau. Former young carer Ngākau explained that she *‘was working in a dairy [shop] after school and all weekend and all that money went as a financial contribution to our household … I was working like 10-h shifts … So I was, in effect, supporting our family from a really young age*.’

Service coordination and translation was carried out by 68% (n = 19) of participants and included accessing, coordinating, translating, and overseeing support services. Former young carer Chun said that a key part of her role for her mother who had medical and neurological conditions was *‘being a broker [by] helping with referrals [and] maintaining all of the medical records … I can’t expect my father to do that because he didn’t really understand the health system and the language would have been a problem … He speaks Chinese.’* Some young carers described the difficulty of translating because of the intimate nature of content being discussed or their concerns to correctly translate such key information.

#### 3.2.2. Dimensions of Emotional, Social, and Spiritual Care

Whilst physical care initially dominated participants’ discussions, interviews soon revealed a further three categories of care integral to young carers’ roles: emotional, social, and spiritual support. All young carers undertook emotional care through providing comfort and a listening ear. Former young carer Mary said that caring for her mother with a medical condition involved *‘always texting mum [to] make sure she was okay, giving her a call at lunchtime … [I] was juggling between work and trying to manage from afar’* (carer age 19–25). Participants whose care recipients had a mental illness described *‘picking up on’* subtle changes to their care recipient’s behaviour to *‘avoid’* or *‘de-escalate’* emotionally fraught scenarios. For many, emotional support also meant holding in or concealing their own emotions. This was the case for former young carer Kahurangi whose sister had a mental illness. She recalled ‘*a situation where [we] had a big family fight. It nearly turned into a punch up! And [my sister] … took some newspaper and set it on fire and said she was going to burn us all down … because she could not cope with the fighting. It made us really aware of our … own emotions … I’m so conscious about keeping it together that now [even when] I want to break out I’m too programmed.’*

Social care was also undertaken by all young carers and included keeping their care recipients company and maintaining their links to their immediate and wider social networks. Former young carer Louise supported her father who had physical and mental conditions: *‘to me, young caring means … being someone to talk with. Like when dad was isolated, being his company and his friendship’* (carer age 5–21). Social care also extended to other whānau members and in particular primary adult carers. Former young carer Wyn said that once her mother became unwell with mental and medical conditions, *‘dad didn’t do anything outside of work and home [so] I’d make sure we’d to go to the [movies] every week’* (carer age 8–14). Social care was particularly commonplace for care recipients who had limited social networks, were largely housebound, or were admitted to outside-of-home services. Former young carer Rachel explained that for her brother with a neurological condition, *‘socially he was with me practically everywhere I went … because I knew if I didn’t do that he [would] have … a very lonely, isolated life’* (carer age 13–21). Māori, Pacific, and Asian young carers enacting a collective whānau model of care highlighted the importance of social care, which *‘united whānau’*. Pacific former young carer Mele, who supported her grandmother who had a mental and physical condition, explained how ‘*all the family functions were at our house ’cause they wanted to see grandma … So that’s not just the nuclear family, it’s the extended … That was the most challenging thing of caring ’cause me and my sisters were basically … washing plates and serving food for the whole day’* (carer age 18–24).

A unique spiritual dimension of care was evident for young carers of all ethnicities and faiths. Participants recited karakia/prayers and sang hymns to maintain their loved ones’ spiritual connectedness. Mele explained that *‘for Pacific cultures there’s a lot of spiritual connections, especially about religion. [So] we always said prayers with [grandma] and sing hymns for her … and you’d see [in] her face—she’d be at peace.’* Some participants spoke about their caregiving roles as part of the bigger picture of caring within their cultural and religious context and within the whole whānau past and present. Former Māori young carer Kahurangi explained that *‘caring is not individual … Even if you might be the only person there, you’ve got all your whānau with you, [and] recognise, your ancestors. So, when you’re doing something it’s never ‘I’. It’s always about ‘we are*’.’

#### 3.2.3. Difficulty Ascertaining the Extent of Care

While the nature of care as encompassing all four elements of wellbeing was clear, the extent of care was difficult to ascertain due to: (1) variance in what ‘counts’ as care; (2) the episodic nature of disability and illness and the tendency to recognise only crisis-related care, especially among young carers of loved one with mental illness and substance misuse; (3) unclear numbers of care recipients being supported; and (4) nonfinite loss underlying care tasks.

Firstly, the extent of young carers’ roles could be difficult to ascertain, with variance between participants’ descriptions of tasks they deemed *‘young caring’* and those they deemed *‘normal’* for their age at the time of providing care. For example, former young carer Terrence initially did not count domestic care as part of his role caring for his mother with mental, medical, and physical conditions when he was in his early twenties; *‘I know how to like wash clothes and make the bed, but I mean [at] my age … it’s like real normal’* (carer age 22–25). However, Louise felt domestic care tasks, undertaken from the age of five years old for her parents with medical, physical, and mental conditions, did count: *‘doing the cooking and the cleaning and all that kind of stuff around the house [was] a big commitment’* (carer age 5–21). Interestingly, care tasks that some participants deemed age-appropriate at the time of providing care were reconceptualised during their interviews with the benefit of hindsight. In these cases, participants recognised considerable challenges underlying traditionally *‘simple’* or *‘age-appropriate’* tasks. For example, housework was initially dismissed by former young carer Dan, who described it as *‘just the basics … I just did the laundry, cooking, cleaning.’* However, Dan later identified the difficulty of enacting such tasks because his mother with a mental illness *‘was in a deluded state continuously … she was a hoarder [so] the house was in a hell of a state; there was two feet of newspaper and hoarded stuff and rotting food in the kitchen … [but] I was still a kid, I could only do what I could.’*

Secondly, the extent of care was hard to determine when changes in care recipients’ health resulted in ‘varied’ and ‘unpredictable’ caring roles for over half (57%, n = 16) of participants. This was especially, but not exclusively, evident for care recipients with a mental illness or substance misuse, whose ‘episodes’ could necessitate periods of intensive care followed by relative calm. Former young carer Kahurangi said that her sister with a mental illness *‘would go from being catatonic to really animated and a little bit psychotic, [and] there would be an episode and then often the police were called because there’d be a violent outburst or she would go missing … So [at those times] we were on edge [and] very conscious of … her wellbeing. That’s quite draining.’* Due to such episodic caring roles, many participants initially underestimated the breadth of care that they provided, often only identifying intensive care provided in response to episodes or peaks in their loved one’s disability or illness, which was not necessarily representative of their ongoing caregiving realities. As such, despite young caring roles continuing in *“down times”* (Lucy) or *“periods when she was very well”* (Dan), care in this period could be overlooked as tasks did not represent the intensity of episodes, peaks, or relapses. Determining the extent of care from the initial half hour of interviews would thus have painted a very different picture to the overall reality of caregiving that emerged as the interviews unfolded.

Thirdly, the difficulty determining the extent of care included challenges in identifying the number of care recipients for whom a young carer was responsible. Six participants began their interviews understanding that they had one care recipient but elaborated on their roles to the point that additional care recipients were identified. Participants’ confusion regarding the number of care recipients they supported could be due to their residence in large two-plus adult households, providing care for a care recipient in formal services, or support of healthy siblings in addition to their unwell loved one(s). Current young carer Phoebe, aged 16, initially discussed caring for her grandmother who had medical and physical conditions, but as her interview progressed and she described moving in with her mother who had a mental illness, an additional care role for her younger brother was identified. Phoebe explained, *‘when [mum] was having her episodes … I played sort of a mum role to my baby brother. He would always come to me when he wanted hugs or when there was something wrong … [mum] didn’t really know how to do the whole ‘caring mum’ thing, so he looked to me for that comfort’* (carer from age 10).

Fourthly, nonfinite loss made the extent of care hard to ascertain wherein a hidden layer of grief underpinned young carers’ roles. Nonfinite loss is the often hidden and enduring grief following a negative event, which leaves an ongoing physical or psychological presence [43]. Reflecting previous studies highlighting the grief experienced by young carers [44], nonfinite loss was evident as young carers described the gap between their and their loved ones’ lives as they had or should have been, and their current realities, with many young carers grieving the loss of their loved ones even before they passed away [45]. Louise commented that after the onset of her mother’s medical condition, *‘it was really hard to have a mum but yet not have a mum. So she was my mum, but she wasn’t anything like how I remembered, and she couldn’t do things that other mums did. I think [caring for mum] was a reminder that things were different then and not how you wanted them.’* Nonfinite loss created a layer of complexity underlining young carers’ roles, as care tasks mirrored the degeneration of their care recipient’s abilities and personality [45]. Terrence explained, *‘the hardest part was definitely seeing how much my mum changed … It’s weird because my mum is there but it’s not my real mum at home; I so miss my mum. I miss her ‘cause I know the person she was, her spirit.’*

### 3.3. No Shared Young Carer Identity

Despite carrying out significant care roles, only four participants (16%) had heard the term young carer prior to their involvement in this study. Most participants only became aware of their young carer identity after watching the first author’s recruitment video outlining her own young caring experience. Former young carer Melanie said that despite two years caring for her friend with a mental illness, *‘before I saw your research I hadn’t ever really considered myself to be a young carer. That was a little bit of a revelation to me. For the first time, I had a label to give that experience [and] it was a good feeling.’* The four participants who had heard of young caring prior to their involvement in the study described *‘discovering’* the term on the internet, via adult carer organisations, or whilst living overseas. These participants felt empowered when they initially became aware of and identified as young carers. Former young carer Amelia, who supported her grandmother and aunt with physical conditions, said; *‘Auntie was listening [to] the [Pacific] radio station and heard about [young caring] … and then from there I kind of feel like I’m almost an important person now, because I’ve been recognised by doing what I do … Whereas before I don’t even know or realise*’ (carer age 6–25). The same positive experience was expressed by the many participants who self-identified as young carers through their participation in this study. These participants explained that identifying would have allowed them to inform other people of their roles, meet other young carers, and receive support. Former young carer Terrence who supported his mother with medical, mental, and physical conditions said, *‘there needs to be more support because honestly, I was burnt. There just wasn’t any assistance for me, and I have a life too! And I was lonely, man.’* However, the four participants who identified as young carers prior to this research stressed that recognition did not result in support and connection, due to the absence of young carer awareness and services in NZ. Current young carer Grace, aged 21, said that identifying during her time caring for her sister with a physical and mental condition *‘didn’t actually mean anything in the end … [it] would have been really cool if I’d met other young carers that were going through the same thing as me, and we could just be like on a mutual understanding of each other’s lives … but there was nothing’* (carer from age 6).

### 3.4. Lack of Support Services Created Further Stress

The widespread lack of support extended beyond young carers themselves to inadequacies in services, communication, and culturally relevant support for care recipients with disabilities or illnesses and their whānau. A key reason for children and youth’s caring roles was the difficulty of accessing appropriate services for their care recipient, which even once accessed were often piecemeal, narrowly focused, inflexible in their delivery, and culturally unsuitable. Many young carers said that whānau were expected to approach services, despite their lack of understanding of the breadth of support available, and without guidance to navigate the process. Former young carer Tilly, who cared for her sister with a physical disability, described how services *‘definitely take advantage of people … because [they’re] not going to tell you if you are entitled to something … You have to really push to check if something is available [and] it’s quite hard as a young person’* (carer age 18–25). In fact, some young carers described a model of service access whereby whānau were required to fight for services to secure the support they were entitled to. Current young carer Leah, aged 17, said that even though her brother with neurological and mental conditions *‘is what you’d call completely dependent’*, accessing services was *‘a battle … Recently, they decided that he was going to be mentally ill and not [neurological condition], which is just like anyone else! … [So] mum’s on the phone for hours at a time negotiating funding’* (carer from age 3). Young carers understood that this fight for services required *‘disabling’* their care recipient, which stood in contention with the dignity and autonomy the whānau worked hard to uphold.

Even after overcoming access hurdles, many young carers described the difficulty of coordinating several narrowly focused services with a continuous stream of new and unfamiliar care staff. As such, most participants felt that no single service or formal carer was able to gain a holistic picture of their care recipient’s health and overarching needs. Former young carer Miharo explained the high turnover of care staff for her mother with a physical and mental condition; *‘I knew she was burning people out because I could hear the caregivers quietly saying, ‘Oh my God, I don’t know if I can keep doing this’ … Then the district nurses stopped coming [so] the responsibility [on us kids] was huge.’* In the two cases where young carers had long-term care staff, there was significant, positive impact for the whānau. Former young carer Louise who supported her parents due to medical, physical, and mental conditions said that her whānau ‘*had a couple of district nurses come in that I got to know really really well over the next eight or nine years … they really cared.’*

Difficulty managing formal care services was exacerbated by a dearth of communication both between the individual service providers and between service providers and the whānau. Such lack of communication often resulted in services that were not responsive to the episodic nature of care recipients’ disabilities or illnesses nor the changing circumstances of the whānau unit in which the care recipient was embedded. So, despite their care recipient being in receipt of services, young carers often filled in service gaps to provide more *‘dynamic’* care. Former young carer Miharo explained that over time her mother’s physical and mental conditions *‘got worse and [so did] the services. I remember [walking into] mum’s room and … she had slit her waterbed to give herself a bath because the district health nurses hadn’t been to see her all week. She had been lying there for three days in her own urine and faeces.’*

Services did not cater to the seven young carers from families who did not speak English as the primary household language, instead requiring young carers to translate during English appointments. Participants not only translated words but also dominant notions of disability, illness, and care underlying service discussions, translating them into culturally relevant understandings and worldviews. For instance, former young carer Chun translated using her father’s understanding of mental health, shaped by his traditional Asian culture, because *‘I didn’t want to frame it like [mental illness] because it means [dad] might just shut off [so I] … framed everything from that sort of Buddhist religious lens.’*

Some services matched formal care staff to whānau based on culture, which the Pacific participants said was inappropriate because they could expose stigmatised illnesses or disabilities to the wider tight-knit community. Former young carer Mele said, ‘*We wouldn’t want [carers from our own culture] because grandma has [mental condition], which is quite a stigmatised disease … We’re a small community in NZ and they would expose our family.’* The importance of recognising variance both between and within cultural communities and individual whānau/aiga/families was highlighted, with a Māori young carer explaining that her whānau would prefer a Māori carer.

Despite describing the inadequacy of formal services, participants were often open to such support, with several young carers encouraging an integrated model of care, uniting whānau and formal services.

### 3.5. Poor Outcomes for Young Carers

Unsurprisingly following the discussion of a lack of support for young carers and their whānau, all young carers experienced some degree of poor outcomes as a consequence of young caring.

#### 3.5.1. Positive Attitudes towards School but Poor Educational Outcomes

Overwhelmingly, young carers and their whānau held positive attitudes towards school, which provided respite, facilitated a life outside of young caring, and offered stability and predictability. Former young carer Lucy who supported her mother who had a medical condition said that *‘Learning has always been a bit of a salvation for me, like a comfortable place … [an] outlet.’* However, most participants’ capacity to engage at school was disrupted by their caring role. Current young carer Phoebe, aged 16, said that due to her mother’s mental illness, her mother would ‘*hang out with quite dodgy people and make really bad decisions … We were always staying at other [people’s] houses after parties, so I had quite a lot of days off school, and I hated it.’* Even when young carers were present in class, tiredness because of providing care alongside concern for their care recipient often impacted their ability to learn. Former young carer Sally who cared for her mother with mental and medical conditions explained; *‘There was all sorts of things that just made it really difficult … to concentrate, to be present, because in the background you’ve got this pressure and worry: Is today the day that I’m going to be called ’cause she’s been successful [in her suicide attempt]?’* (carer age 11–25). Most participants also described being unable to complete homework, meet deadlines, and prepare for examinations due to their caring roles. As such, young carers often understood that their school performance was not reflective of their actual ability. Former young carer Louise who supported both her parents with medical, physical, and mental conditions, said that *‘Most kids would just go home [after school] and all they have to do is schoolwork, but it wasn’t really a priority for me … In my final years [of school] I was picking easier subjects just because they took less effort; I didn’t have to do so much at home for them.’*

Caregiving interrupted young carers’ ability to participate in extracurricular activities despite their desire to do so. Most participants joined an elective activity, but almost all young carers reported the challenge of seeing their commitment to fruition. Current young carer Atalanta, aged 12, who supported her mother due to a medical condition, said that although she was a member of a school band at the time of her interview, *‘it’s annoying ’cause like, if I say [to the band] ‘I need to be back by this time’, they’ll be like, ‘No ’cause you’ve got commitments’, and I’m like, ‘Well, being home for mum is my commitment’* (carer from age 7). Several young carers who successfully maintained an extracurricular activity said that they cherished the opportunity to receive respite, develop friendships, gain a sense of achievement, and take on leadership opportunities. Former young carer Dan, who supported his mother with mental illness, described how playing sports ‘*just really gave me a lot of strength … it meant the world to me.’*

A significant finding is that no young carers disclosed their caregiving roles to their teachers because participants did not want to be perceived as *‘different’* or did not believe their teachers would understand their roles. Referring to caring for her mother with a medical condition, former young carer Lucy asserted; *‘There was no way that [my teachers] could have understood what I was doing.’* However, several young carers who disclosed aspects of their loved one’s disability or illness to their teachers—but in no cases their caring roles—spoke about the positive impact of teacher awareness on their education. Current young carer Leah, aged 17, said that talking to her deputy principal about her brother’s neurological and mental conditions and the difficulties that she was having at school resulted in *‘dispensation from homework if I need it … which was great! I think my grades have improved because I’m not sitting there stressing about 10 million different things.’*

#### 3.5.2. Peer Socialising Outcomes

Non-disclosure of their caring roles extended beyond their teachers and was a key reason underlying participants’ struggle to form deep and meaningful relationships with their peers, alongside a lack of shared life experiences and limited time outside of school hours to cement friendships. In some instances, young carers intentionally resisted close friendships by selecting peer groups who would not be interested in their home lives. Current young carer Grace, aged 21, who supported her sister with physical and mental conditions, said that she intentionally ‘*hung out with the wrong crowd … I didn’t want to let anyone in to know [the young caring] side of me … [And] those sort of people will only attach themselves to you for a good time out, but there’s no close bond.’* Nevertheless, the importance of deep friendships was highlighted by the seven young carers who succeeded in forging strong connections. During his final high school years, former young carer Greg cared for his father with a medical condition. He explained how much he valued his *‘friend who had lost her father … We bonded over that … [because] the realities of terminal illness and death and painkillers and hospital [are] just concepts that are entirely foreign to other people of this age bracket. They were totally naïve’* (carer age 17–21).

Worryingly, over a third of young carers reported being bullied. In all cases their peers were unaware of their young carer roles, and instead participants believed that their heightened maturity, their peers’ awareness of their loved one’s disability or illness, or their whānau’s difference (e.g., as a single-parent household experiencing poverty), led to bullying. Current young carer Leah said that she *‘didn’t like intermediate [school]. I got quite badly bullied there. Part of it was to do with [my brother] being disabled … I think it probably started with me being completely naïve and assuming that everyone not only found it normal but would understand and not think it was weird.’*

#### 3.5.3. Early School Exit and Not in Education, Employment, or Training (NEET)

Over 20% (n = 4) of the 19 young carers who provided care whilst of secondary school age (12–17) were early school leavers, exiting school prior to the start of Year 12 (aged 16 years and under). The four early-exiting young carers said that their premature departure from school was due in part to a lack of support to maintain both school and caregiving. Current young carer Grace, 21, who cared for her sister with physical and mental conditions explained that she *‘left school really young, 15 [years old] … I was miserable … No one knew what [I was] really going through … that I’m so fucking busy at home … They just saw me with shitty grades, where I would just wag classes, forge notes. They just saw me as a badass, but … as much as it didn’t look like it, I just wanted help.’* Furthermore, almost half of participants who provided care whilst of secondary school age considered leaving school early but did not due to having *‘one really great’*, *‘supportive’*, or *‘really influential’* teacher or coach (42%, n = 8). Former young carer Lucy, who cared for her mother with a medical condition, explained how she ‘*just didn’t think it was a possibility [to stay in school], but luckily I had this teacher who … just in the background was being an advocate … She was incredibly supportive.’*

Whether young carers left school early or not, most participants described feeling *‘stuck’* with regards to their higher education and employment opportunities, often due to uncertainty surrounding the length and intensity of caregiving, which resulted in a lack of future-planning opportunities. Current young carer Phoebe, aged 16, who cared for her grandmother with a medical and physical condition and for her brother due to her mother’s mental illness, said ‘*it feels like I’m putting things on hold for me … I worry more about what my nan and my brother and my mum need, rather than what I need … they always come first before my own stuff.’* As such, young carers were not only planning their young adulthood around their current caring role, but many were also factoring in possible future care. Former young carer Chun explained ‘*that with my mum’s disability [progressing], I will have to take that [full-time] role on again within the next five to 10 years … my life will go on hold … so it makes me kind of anxious … What if my [study] didn’t get finished before that happens? Or what if I want to do something else after the [study] and I run out of time?’*

Participants’ lack of ability to plan could translate into their experiences of being NEET. Over 50% (n = 12) of participants were NEET for at least one year between the ages of 16 and 25. When young carers did attend university or training they found it challenging to fully commit, so whilst eight participants began university or training, 75% (n = 6) left their studies prior to completion, due to their caregiving roles. Former young carer Lucy said that due to her role caring for her mother with a medical condition, ‘*I dropped out [of university] after like two weeks, ’cause I couldn’t do both: it was impossible. Which started a long career of dropping out of uni. So … I was just looking after my mum, which was quite intense.’* In addition, many young carers felt that their employment opportunities were limited and did not reflect their areas of interest or expertise due to their competing caregiving roles. Current young carer Grace said that alongside caring for her sister with a physical and mental condition, ‘*I learnt how to make coffee … it was enough to get a foot in the door for hospo [hospitality] work … [But] I’m now getting shitty pay and slogging my guts out.’*

#### 3.5.4. Negative Physical and Mental Health Impacts

Many young carers experienced physical injuries due to the difficulty of lifting or moving their care recipient, the stress of providing care, and/or a lack of self-care. Former young carer Terrence described the impact of caring for his mother with medical, mental, and physical conditions; *‘I put on over 100 kg … I think I was too busy taking care and being there for everyone else that I didn’t pay attention to myself for anything.’* However, it was the mental health impacts of young caring that were most common, with over 60% (n = 18) of young carers experiencing anxiety and/or depression during and after their time providing care. Former young carer Chun who supported her mother due to medical and neurological conditions said that during her time caring, she *‘would feel pain in the chest because [of] all my anxiety … you can’t breathe and you think you are dying.’* Despite experiencing poor mental health*,* most young carers *‘didn’t tell anyone’* how they were feeling, especially when their care recipients were their only source of support, or when their parents or caregivers were the primary adult carers. Former young carer Rachel who supported her brother who had a neurological condition explained, *‘the older you get the more you see how much your parents carry with a disabled child in the family … and so rightly or wrongly, one way of relieving that burden is to not share your own stuff with your parents.’* In lieu of having someone to talk to about their poor mental health, participants often self-medicated with drugs and alcohol or coped using self-harm. Former young carer Lucy said that she had *‘fairly unhealthy coping mechanisms’* while caring for her mother with a medical condition, explaining ‘*I was cutting myself … because it felt quite nice.’*

Overall, participants were carrying out a range of care tasks, many without awareness of their young carer identity, and in the absence of suitable formal support. As a result, all young carers experienced a degree of poor physical and mental health outcomes.

## 4. Discussion

This study revealed that many of the NZ young carers enacted natural, holistic care to meet their care recipients’ physical, emotional, social, and spiritual wellbeing, and conceptualised their roles as being embedded within the wider functioning of their whānau unit. However, the lack of appropriate services resulted in many undertaking overwhelming roles that no longer reflected a natural whānau conceptualisation of care.

Overwhelming care roles. There is sustained debate in the literature regarding whether children and youth should be involved in caregiving roles at all [46,47,48]. Participants’ conceptualisations of their roles as an integral part of growing up in NZ adds a key consideration to this debate. Their desire to care is supported by the commitment in Article Two of NZ’s founding document, Te Tiriti o Waitangi [49], which protects the rights of Māori to live as Māori [50], and by Article 30 of the United Nations Convention on the Rights of the Child (UNCROC) [51] to enjoy and enact their whānau and cultural customs. However, an overwhelming care role could instead compromise children’s rights.

Reflecting the well-documented “caregiver burden” [52] (p. 5) and overwhelm [21,53] attached to young carers’ roles, the results of this research identified that it was the overwhelming nature of caring—rather than simply children carrying out care—that requires attention. Such an overwhelming model of care, largely brought about by inadequate services and lack of support for care recipients and their whānau, resulted in negative impacts, including poor educational outcomes, limited opportunities to socialise, difficulties attending higher education, and poor mental health.

Misalignment between experience and definition. Participants’ understanding of young caring as a natural part supporting their loved one’s holistic wellbeing is in contention with the prevailing understanding wherein young caring involves substantial or significant care [18,22]. Furthermore, existing definitions of young caring may unintentionally exclude diverse cultural conceptualisations of whānau and childhood by prioritising a Global Minority framing, evident in young carer’s roles being described as parentified [54], associated with an adult’s role [18,23], or a “role reversal” of child and adult responsibilities [55] (p. 77). Existing studies highlight that young carers who perceive their roles as culturally “normal” [16] (p. 15) or “natural” part of being in a family [52] (p. 6) can be particularly hidden and their voices missed in research. Webber asserted that vulnerable populations are often “exposed to research that is driven by dominant worldviews … that can exacerbate their vulnerability” [1] (p. 129). Phenomenographic interviews were central to empowering the participants to choose the parameters of young caring that held meaning and to unravel their experiences in a way that felt natural to them. As such, participants’ own understandings and ways of knowing regarding being a young carer led the interviews, rather than researcher-led questions that could limit the meanings of their experiences.

Redefining young caring is imperative so that research, policy, and services capture diverse NZ young carer experiences. In line with international research that broadens the young carer definition to account for diverse young carer experiences [56], an alternate definition capturing NZ young carers’ experiences is proposed:


*NZ young carers are children, youth, and young adults aged 25 years and under who physically, emotionally, socially, and/or spiritually support loved ones due to ill health, disability, substance misuse, mental illness, advanced age and/or socio-economic conditions. Such support can be enacted within or outside the home, alone or as part of a wider whānau/aiga/family or friendship unit, provided to the unwell individual or another person/child, and undertaken continuously or intermittently.*


Participants’ conceptions of the normalcy of care in their whānau, the absence of a shared NZ young caring identity, and a widespread societal-level lack of awareness of young caring suggest that the number of NZ young carers may be larger than estimated, and many may be unaware of or unsupported in their roles.

Limitations. Several possible constraints emerged during the study. Firstly, the first author’s own experiences as a young carer could have influenced how she interpreted the findings. To mitigate this, she engaged in autoethnographic journaling and regular discussions with her supervisors to reflect on and explore underlying biases and justify emerging categories of description. Secondly, the extent of the emotional impact of interviews on the first author was unexpected. Regular counselling provided an outlet to discuss her emotions and develop strategies to manage them. Thirdly, interpretations related to culture may reflect the first author’s identity as Pākehā with limited knowledge of Māori, Pacific, and Asian cultures. As such, contextualisation of the responses of such participants derived primarily from extensive reading about cultural perspectives on care, disability and illness, childhood, and whānau. Fourthly, the non-random sample may have limited participants to young carers with access to Facebook via which the video was circulated and/or the NZ carers’ magazine [36]. Young carers without such access could have been excluded, although a number were told about the research by an extended family member or friend. Mitigating this potential limitation is research that shows that social media use is “ubiquitous” among NZ youth aged 9–17 for socialising and entertainment, and the high percentage (90%) who regularly watch online videos [57] (p. 1). In addition, this innovative approach produced the largest cohort of NZ interviewees to date.

## 5. Potential Impact on Policy, Research, and Service Provision

Drawing from young carers’ experiences and needs shared in this research, we present key recommendations for young carer policy, research, and services in NZ that should be based on a broad definition of young caring.

Policy. The single most impactful action NZ can take to recognise and meet the needs of young carers is to implement the four young carer Actions in the 2019–2023 Carers’ Strategy Action Plan [5], which have the scope to support all recommendations outlined in this section. Given the tendency for decision-makers to unintentionally exclude diverse cultural conceptualisations of young caring [1,18,23], the Action Plan should be implemented via culturally responsive practices that reflect the children and youth it is designed to support.

Research. Achieving balance for young carers under the UNCROC and respect for cultural conceptualisations of childhood, whānau, and care outlined in Te Tiriti o Waitangi necessitates providing young carers with the opportunity to be heard [51] and for their opinions to be considered in decisions affecting them [58,59]. We recommend targeted research to capture the experiences of more young carers who are Māori, Pacific and/or represent NZ’s increasingly diverse population. Such research can deepen and inform existing international studies of the experiences of young carers from Black, Asian, minority, and ethnic (BAME) [60] and culturally and linguistically diverse (CALD) [61] backgrounds.

We also recommend targeted research with children and youth supporting loved ones with mental illness or substance misuse, who appeared to have particularly challenging caregiving roles due to the episodic nature of their care recipients’ ill health and the lack of mental health supports. However, the finding that the majority of participants were providing care across multiple categories of illness or disability means that more research is needed to further understand the specific impacts related to mental health or substance abuse. Such research will make important contributions to the literature regarding children of parents with mental illness (COPMI) [62].

Such targeted research will be essential to continue refining our understanding of young carers’ experiences, so that policy and services truly reflect their realities and needs. It is imperative that such research utilises a broad definition of young caring, as defined in this paper and increasingly adopted in international literature [56], and recruitment methods that support young carers to self-identify via storytelling and shared experience.

Services. The unwillingness or inability of young carers and/or their whānau to access services shows the need to review the current model of service delivery which they identified as difficult to access, piecemeal, narrowly focused, inflexible, and/or culturally unsuitable. It could be advantageous to explore the existing Whole Family Approach utilised in the UK [20] which enables a holistic understanding of an individual’s support needs and considers their wider support network. Assigning a coordinator or advocate with in-depth knowledge of NZ services to each young carer’s whānau would help them navigate an unfamiliar and confusing services landscape and rebuild trust in services.

We also recommend awareness-raising at a societal level that highlights the holistic nature of young carers’ roles and the challenges they face and is framed by positive, mana(esteem)-enhancing narratives. The finding that young carers are unlikely to reach out for support shows the necessity of education and professional development for people who are most likely to encounter them, including school and university teachers and counsellors and service providers. Awareness-raising should be accompanied by targeted young carer support comprising opportunities for young carers to connect with one another, receive mental health support, attend respite activities and events, and be supported to manage the demands of caring alongside school, university, and/or work.

## 6. Conclusions

In conclusion, this phenomenological research revealed the complex lives of young carers in NZ and the pleasures and challenges of their roles. The findings reinforce existing research about the multi-dimensional nature of caring, including physical, emotional, and social care. This research also revealed a more hidden dimension of care being spiritual support, which was provided by carers of all ethnicities. In addition, the research identified the overwhelming nature of care roles and uncovered the depth of nonfinite loss that infused their experiences, as young carers grieved the loss of their prior or expected relationships with their care recipient. It also exposed important gaps in support for young carers and reinforced the need for responsive, wrap-around support services that include the whole whānau without marginalising the young carer, many of whom started caring at very young ages.

Overall, the research indicates that in order to improve the lives and outcomes for young carers and their whānau, NZ must translate research and policy into recognition and provision of support that meets their unmet needs. It is vital to shift the dominant narrative to one of autonomous, supported young carers carrying out their natural familial and culturally informed roles, with resources and within a context that supports the current and future goals of the young carers, their care recipients, and their wider whānau. The most important catalyst for change will be prioritising their voices in the policy arena, and implementing services that support their success both within and outside of their caring roles. We cannot delay recognition and support for NZ young carers any longer: it is time to talk about, and deliver on, the expressed needs of NZ young carers. As young carer Lucy explained: *‘if we don’t talk about young carers then we continue not to service their needs. And the most important thing is to have our voices heard: to make silent voices louder … Sharing our stories is the most important way to be heard.’*

## Data Availability

The data presented in this study are available on request from the corresponding author. The data are not publicly available due to privacy and ethical reasons.

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
