# Peer review of "Voices from the Margins: Aotearoa/New Zealand Young Carers Reflect on Their Experiences"

_ijerph, 2023, doi:10.3390/ijerph20156511_

Round 1

Reviewer 1 Report

Thank you very much for allowing me to review your paper. It addresses a topic that I am familiar with in my own research activity. I admire your recruitment efforts and I am pleased to see you were, after all the effort, able to recruit young people who were willing to share their experiences.

I have a few suggestions:

1. Although I am aware of the need to preserve anonymity I would have liked to see a little bit more information regarding who were the 25 young people that took part. In terms of age, gender, ethnicity, family characteristics, time since caring responsibilities started, family member they care for (what illness)? This would allow the reader to have a better picture of your sample. 

2. I would like a little bit more detail regarding the data analysis that you carried out. You do provide information from line 143 onwards but I feel that this could be expanded and iterative steps described. 

3. Maybe a table presenting the themes and sub-categories would improve the presentation of the findings. 

4. Some of the 'themes' look to me more like a list (e.g. Care Tasks Undertaken). I think a further in-depth analysis (moving from description to 'understanding/mechanisms') could enhance the outcomes by 'going deeper' into the young people's experiences. 

5. Did you ask young people their opinions about the policies and actions plans that you mention in your introduction section? It would be interesting to know what they think about them and whether they feel represented. 

6. Did you find significant differences between young people who care for a family member with a physical disability/mental health condition? Evidence shows that those two groups present very different outcomes (and that the impact of 'caring' is also different and affect diverse areas). 

It think the quality of the English is good. However, I suggest a detailed re-read and edit throughout. 

Author Response

Thank you so much for the time you have taken to review our manuscript, and for your invaluable feedback. Your comments were easy to navigate, understand, and address, and that is much appreciated. The changes we have made based on your feedback have strengthened the manuscript significantly: Y

You asked for elaboration to ensure the reader has a better depth of understanding of the experiences of the young carers, recommended a table to add further key information, and pinpointed key findings that were missing and needed to add to the overall picture of young caring in Aotearoa/New Zealand. These changes have been actioned - please see the attachment.

Your expertise in young caring shines through in your reviewer comments and we are so grateful to have had your input into our manuscript. Please see the attached table with a point-by-point response to your comments and a description of how they have been addressed.

Ngā mihi nui (best wishes), Lauren, Jan, and Toni.

Author Response

Thank you so much for the time you have taken to review our manuscript, and for your invaluable feedback. Reading your opening comments regarding the importance of the research and richness of the data were very encouraging. We can see the time you have taken to reflect on the paper and carefully consider amendments to strengthen the paper.

The changes we have made based on your feedback add clarity and greater insight into young caring in Aotearoa/New Zealand and how we ‘sit’ within the international context.

Please see the attached table with a point-by-point response to your comments and a description of how they have been addressed.

Ngā mihi nui (best wishes), Lauren Donnan, Janet Gaffney, and Toni Bruce.

Round 2

Reviewer 2 Report

Dear Authors, 

Thank you for your careful and substantial edits to the paper, which I believe have improved it greatly and thus its potential dissemination and impact to policy. A couple minor comments below. 

Thank you. 

Thank you for clarifying the reason for non-random sampling. I do believe there should be a note in the limitations, however, about potential biases of this approach (e.g., what type of young carers are unlikely to learn about the study or to volunteer). In the U.S. context, members of indigenous groups and African-Americans are much less likely to volunteer for research because of a long history of being mistreated in research studies. 

Thank you for adding the table describing individual characteristics, which I would put in the Appendix. I still urge the authors to table to summarize the sample (e.g., percent female). Also, please remember this is an international audience. I read “Pākehā” as a potential indigenous group I was not familiar with. 
